# The Impact of Social Determinants of Health on Metabolic Dysfunction-Associated Steatotic Liver Disease Among Adults in the United States

**DOI:** 10.3390/jcm14155484

**Published:** 2025-08-04

**Authors:** Vidhi Singh, Susan Cheng, Amanda Velazquez, Hirsh D. Trivedi, Alan C. Kwan

**Affiliations:** 1David Geffen School of Medicine at University of California, Los Angeles, CA 90095, USA; vidhisingh@mednet.ucla.edu; 2Smidt Heart Institute, Cedars Sinai Medical Center, Los Angeles, CA 90048, USA; susan.cheng@cshs.org; 3Jim and Eleanor Randall Department of Surgery, Cedars Sinai Medical Center, Los Angeles, CA 90048, USA; amanda.velazquez@cshs.org; 4Department of Internal Medicine, The University of Texas at Austin Medical Center, Austin, TX 78701, USA; hirsh.trivedi@austin.utexas.edu

**Keywords:** metabolic dysfunction-associated steatotic liver disease, MASLD, metabolic syndrome, social determinants of health, liver disease, SDOH, fatty liver, metabolic syndrome

## Abstract

**Background/Objectives**: Metabolic dysfunction-associated steatotic liver disease (MASLD) is a leading cause of chronic liver disease. It has known multifactorial pathophysiology, but the impact of social determinants of health (SDOH) on the rising prevalence of MASLD is poorly understood. We conducted a retrospective cross-sectional study to examine the influence of SDOH on MASLD using nationwide data from the 2017–2018 National Health and Nutrition Examination Survey (NHANES) study. **Methods**: We identified participants with MASLD based on liver ultrasound-based controlled attenuation parameter measurements consistent with diagnostic guidelines. We then used logistic regression models to examine associations between SDOH variables and MASLD, with a pre-specified focus on education and income, sequentially adjusting for sociodemographic factors, medical comorbidities, and other SDOH. **Results**: Our study found that higher education (odds ratio [OR] 0.77, 95% confidence interval [CI] 0.62–0.97, *p* = 0.024) but not higher income (OR 1.12, 95% CI 0.91–1.37, *p* = 0.3) was associated with lower odds of MASLD in multivariable adjusted models. We also identified a significant interaction between education level and food security, as well as interactions between food security and other significant SDOH. In the stratified analyses, higher education was significantly associated with lower odds of MASLD among participants with food security (OR 0.71, 95% CI 0.55–0.91, *p* = 0.007) but not among those with food insecurity (OR 1.26, 95% CI 0.76–2.11, *p* = 0.4). **Conclusions**: Our findings identify the potential impact of SDOH on odds of MASLD and suggest increased importance of food security relative to other SDOH.

## 1. Introduction

Metabolic dysfunction-associated steatotic liver disease (MASLD) has become one of the leading causes of chronic liver disease. It has a global prevalence of 38% among all adults and up to 60% among adults with type 2 diabetes mellitus (DM) [1]. MASLD has multifactorial pathophysiology that includes complex genetic, biological, and cardiometabolic factors. MASLD is characterized by fat deposition in the liver and at least one cardiometabolic risk factor, and over time, steatosis can lead to inflammation and subsequent fibrosis. However, social determinants of health (SDOH) may also contribute to the rising prevalence of MASLD, though their impacts are poorly understood [2,3]. SDOH are non-medical factors influencing health and well-being, which include education level, economic stability, healthcare access and quality, neighborhood and built environment, and social and community context [4,5,6]. Previous studies have explored the effect of diet and physical activity on MASLD [7], but few have focused on upstream SDOH variables like education and income. We aimed to study the associations between SDOH and MASLD in adults sampled in the 2017–2018 National Health and Nutrition Examination Survey (NHANES) study.

## 2. Materials and Methods

### 2.1. Study Cohort

We conducted a retrospective cross-sectional study and included data from the 2017–2018 NHANES study, available at https://www.cdc.gov/nchs/nhanes/. NHANES is a national survey that collects data on adults and children through laboratory tests, examinations, and interviews that capture health, diet, personal, social, and economic characteristics. Starting in the 2017–2018 examination, ultrasound-based transient elastography with a controlled attenuation parameter (CAP) of the liver was included as part of the examination, which allows for estimation of liver steatosis and fibrosis. The NHANES was conducted with approval by the National Center for Health Statistics Ethics Review Board and obtained informed written consent from all the individuals involved in the study. The NHANES 2017–2018 surveys received ethics approval under NCHS Research Ethics Review Board Protocols #2011–17 and #2018–01.

For our study of SDOH and MASLD, we included non-pregnant adult NHANES participants who were aged 20 years and older who underwent ultrasound transient elastography with CAP measurements. We classified participants as with and without MASLD, with MASLD being defined as CAP ≥ 248 db/M with metabolic dysfunction based on American Association for the Study of Liver Disease diagnostic guidelines [8,9,10]. We excluded any participants in the MASLD and non-MASLD cohorts that were missing data on social determinants of health, relevant laboratory values, physical exam findings, or past medical history. The study inclusion flow is shown in Figure 1.

### 2.2. Predictor and Outcome Variables and Co-Variates

Our primary predictor SDOH variables were income bracket and education level, and the primary outcome variable was MASLD as defined above. Income and education were selected among the breadth of SDOH variables available in the NHANES due to the previously established contribution of these specific SDOH on health outcomes [11]. Our covariates included potential confounding factors based on the existing literature [4,12,13]. These included age, sex, race, and metabolic conditions such as patient-reported history of hypertension (HTN), hyperlipidemia (HLD), DM and measured body mass index (BMI) category (i.e., underweight, normal weight, overweight, and obese), and hepatic risk factors including alcohol use frequency and history of hepatitis B or C. We also adjusted for SDOH-based covariates including type of insurance coverage (private versus not private), level of food security (high food security versus marginal or low food security), healthcare access (access to at least one health facility versus no facilities), level of physical activity (moderate versus less than moderate physical activity), and marital status (married/living with a partner versus single, separated, divorced, never married, and widowed), given the potential interplay between these primary predictor variables and the outcome.

Based on exploratory analyses of the association between MASLD prevalence and available education categories (less than high school, high school or GED, less than college/associate’s degree, or college and above, Appendix A), we binarized education level outcome into less than college and college or above, given the clear breakpoint seen in the data. In contrast, there was no clear appropriate cutoff across income levels for MASLD prevalence (Appendix A). Consequently, we binarized income based on the median household income in the United States (US) for 2017–2018, which ranged from $60,366–$63,179 [14]. Because the NHANES reports income in pre-defined income brackets, we binarized income into less than $65,000 and greater than or equal to $65,000.

### 2.3. Statistical Analysis

We conducted our statistical analysis using R/R Studio (version 2024.09.1 + 394). For descriptive analysis, continuous variables were summarized as means with standard deviations, and categorical variables as counts with percentages. To examine the relationship between education level, income bracket, and MASLD prevalence, we performed univariable- and multivariable-adjusted logistic regression. We analyzed the data using three primary models predicting MASLD: education level alone, income bracket alone, and combined education level and income bracket. Each model was sequentially adjusted in three iterations, following the established literature [10]: sociodemographic factors, sociodemographic factors and medical comorbidities, and sociodemographic factors, medical comorbidities, and SDOH variables. Logistic regression analyses were conducted using the gtsummary package and the tbl_uvregression function. We utilized collinearity analysis based on the variance inflation factor to ensure that the included co-variates were not collinear given known potential associations within cardiometabolic risk factors and SDOH. Finally, we tested for interactions between education level, income bracket, and significant covariates. A pre-specified two-sided *p*-value < 0.05 was considered statistically significant.

## 3. Results

### 3.1. Participant Characteristics

We included *n* = 3202 participants and stratified by MASLD (*n* = 1892) and non-MASLD (*n* = 1310) (Table 1).

Among all participants, the mean age was 50.6 ± 17.2 years and 51.3% were male. Participants with MASLD had a greater percentage of participants reporting a history of HTN, HLD, and DM. Among those with MASLD, 71.4% had grade S3 steatosis (>66% liver fat accumulation), and most had minimal liver fibrosis (F0 to F1, no scarring or minimal scarring). In contrast, *n* = 34 participants without MASLD had steatosis grade S1–S3 (>33% steatosis) but did not meet the diagnostic criteria for MASLD due to lack of comorbid conditions. Among SDOH variables, participants with MASLD were less likely to have obtained a college education or above (22.2% vs. 27.9%, *p* < 0.001) (Table 2).

They were also more likely to be married or living with a partner, experience food insecurity, engage in less than moderate recreational physical activity, and report access to healthcare facilities. There were no significant differences in income level between participants in the MASLD and non-MASLD groups.

### 3.2. Primary Analysis: Univariable and Multivariable Regression

The education-level-only logistic regression models showed that an education level of college or above was independently associated with lower odds of MASLD across the three adjusted models (demographics only; demographics and medical comorbidities; demographics, medical comorbidities, and other SDOH, odds ratio (OR) 0.8, *p* = 0.04 in the final model, Appendix A). In contrast, income of $65,000 or more was not associated with odds of MASLD in any model (Appendix A, *p* > 0.05 in all). In the models including both education level and income bracket, patterns were similar across stepwise models (Appendix A). In the final fully adjusted model, an education level of college or above was an independent predictor of lower odds of MASLD (OR: 0.77, 95% CI: 0.62–0.97, *p* = 0.024), but an income bracket of $65,000 or more was not (OR: 1.12, 95% CI: 0.91–1.37, *p* = 0.3) (Table 3).

Similarly to the income- and education-only models, increasing age, male gender, Mexican American race/identity, Non-Hispanic Asian race/identity, history of HTN, history of DM, daily alcohol use, BMI ≥ 25 kg/m^2^, and being married or living with a partner were associated with significantly higher odds of developing MASLD. In contrast, higher physical activity, having full food security, and identifying as Non-Hispanic Black were associated with significantly lower odds of developing MASLD.

### 3.3. Secondary Analyses: Interactions Analysis

We identified a significant interaction between the education and food security variables (OR: 0.59, 95% CI: 0.30–0.87, *p* = 0.014), and thus we performed sub-analysis of the fully adjusted models stratified by food security. In the stratified analyses, higher education was significantly associated with lower odds of MASLD among participants with food security (OR: 0.71, 95% CI: 0.55–0.91, *p* = 0.007) but not among those with food insecurity (OR = 1.26, 95% CI: 0.76–2.11, *p* = 0.4) (Table 4). Within the food-secure group, moderate physical activity was associated with reduced odds of MASLD, while being married or living with a partner was associated with increased odds. However, these associations were not significant in the food-insecure group. No significant interactions were detected between primary predictors and other covariates.

## 4. Discussion

The main findings of our study were threefold. First, higher education level but not income level was associated with reduced odds of MASLD, even after adjusting for demographics, medical comorbidities, and other SDOH. Second, we observed a significant interaction between education and food security, with an education level of college or above being significantly associated with lower odds of developing MASLD only among participants with full food security. Interestingly, among participants with less than full food security, neither education nor any other SDOH variables were associated with odds of MASLD, suggesting that food insecurity may be a dominant SDOH risk factor that supersedes the impact of other SDOH variables. Finally, our findings were consistent with the prior literature in identifying that physical activity, non-Hispanic Black race and ethnicity, and food security are associated with lower MASLD risk, whereas Mexican American ethnicity is associated with increased risk [15,16,17]. However, we also found that non-Hispanic Asian race and ethnicity and marital status are associated with increased risk for MASLD, associations which have been less described in previous studies.

There are several potential explanations for our finding that higher education is associated with reduced odds of MASLD. Firstly, increased education has been associated with improved MASLD self-management, as patients are more likely to have greater access to health information and adopt healthy lifestyle practices [18]. Patients with higher education are also more likely to be recruited for new lifestyle treatment programs and trials compared to people with less education, who may have limited awareness and receive less patient education surrounding the connection between healthy diet and physical activity and lower chronic disease risk [16,19,20,21]. However, notwithstanding the persistent independent effect of education on odds of MASLD after adjustment for food security, our findings suggest significant effect modification by food security shared across SDOH variables. Previous studies that have explored risks for non-alcoholic fatty liver disease identified that the protective effects of higher education were mediated through high-quality diet and physical activity [7,16]. This is supported by dietary data demonstrating a 2.47-fold increase in the risk of liver steatosis for obesogenic diets compared to fiber-based and lean-meat-rich diets, especially the Mediterranean diet [22]. In fact, the Mediterranean diet is a proven treatment for MASLD, leading to a 23% reduced risk of disease, thought to be mediated by its anti-inflammatory and anti-oxidative effects [23]. Furthermore, studies evaluating the effect of time-restricted eating in patients with MASLD have shown improvements in liver stiffness and steatosis [24]. We hypothesize that food-insecure households have an inability to consume a high-quality diet and thereby do not yield the benefits of high educational levels. Interestingly, the elimination of significant effects from the physical activity and marriage status SDOH markers may suggest either (a) dietary drivers of MASLD are particularly strong, and outweigh beneficial effects of education or physical activity, or (b) food insecurity itself represents latent confounders which, even after adjustment for clinical risk factors, demographics, and other markers of SDOH, are particularly deleterious in terms of odds of MASLD. Previous studies in Hispanic populations have shown a strong association between hepatic steatosis and food insecurity, likely due to dietary habits [25]. Either way, our research suggests that targeting food-insecure populations may be a particularly efficient direction of intervention and study.

The education-level findings are in contrast with the observation that individual income was not significantly associated with odds of MASLD. While somewhat surprising, as one may expect that higher income may enable better access to healthier foods and lifestyle change opportunities, this may be explained by the geographic diversity of our sample resulting in pre-defined income brackets providing an inaccurate representation of actual income given different geographic cost-of-living levels. Previous research has suggested that income influences MASLD risk through factors such as food security, access to physical activity, and the presence of medical comorbidities [26,27]. The effect of individual income is markedly reduced after adjusting for education because education may be a better predictor for social factors like neighborhood and physical environment [28]. However, our research did not show a significant effect of income even in unadjusted analyses, suggesting against this being an explanation in our study. While it is possible that our household income cutoff of $65,000, which was based on median US income in 2017–2018, may have been too low to detect an independent effect on MASLD, the non-linear pattern of association suggests that at least in our population, the lack of significance is a true finding.

We also identified that marital status, physical activity, and race and ethnicity were associated with MASLD. This is in line with previous research, which has demonstrated increased moderate physical activity to be associated with reduced risk for MASLD [15,29,30]; racial differences, including the increased risk of disease in Hispanic populations and the lower risk observed in Black populations, may be related to a polymorphism in the PNPLA3 gene, which promotes hepatic fat accumulation [31]. Interestingly, while data on MASLD rates in Asian-American individuals relative to other demographic groups are limited [32,33], studies conducted in Asia have found that 13–19% of Asian individuals also have a polymorphism in the PNPLA3 gene, which may help explain our finding of increased MASLD risk in the Asian subgroup after adjusting for medical co-morbidities and SDOH variables [34]. The Asian subgroup population is also afflicted with MASLD at lower BMI values, a condition called ‘lean MASLD’. Finally, our finding that married status appears to be associated with increased risk of MASLD has been observed in other populations [35], but clear mechanisms for this are uncertain. It is also known that odds of obesity, a risk factor for MASLD, are 88% higher among married individuals compared to single, divorced, and widowed individuals [36], for which the reasoning remains to be elucidated. Previous studies observing similarities in the cardiometabolic profiles of married couples suggest that these consistencies are driven by shared environmental and behavioral factors [37].

Several limitations of the study merit consideration. Firstly, given data availability limitations in the NHANES and collinearity of variables involved in MASLD diagnosis, our analysis is unable to account for all contributing factors, such as anthropomorphic measurements, dietary patterns and intake, medication use, and frequency and type of physical activity. Furthermore, NHANES food security data are collected with the US Food Security Survey Model, and it captures household and not individual food security. As a result, food security may not accurately reflect an individual participant’s risk for developing MASLD. We did not exclude alcohol use, thus potentially combining the MASLD and metabolic dysfunction and alcohol-related (MetALD) categories [9]. Unfortunately, NHANES only reports data on drinking frequency and average number of drinks consumed per day, making it difficult to isolate which participants engaged in significant alcohol use. Sensitivity analyses show that when we remove participants who drink alcohol daily and weekly, assuming this group meets the MetALD criteria, the results remain consistent across all logistic regression models. The cross-sectional design limits the ability to assess longitudinal changes in income, timing of education, and duration of MASLD. The dichotomization of our variables may also not fully reflect some of the subtleties across different groups, which may have affected our income-based analyses. In sensitivity analyses, we tested income as a categorical variable and evaluated different income bracket cut-offs, and the results remained consisted across logistic regression models. Survey designs have specific limitations, including recall and social desirability bias from the participants. Finally, the cross-sectional nature of the NHANES data prevents causal inference.

## 5. Conclusions

In conclusion, we found that education level but not income bracket is associated with odds of MASLD. We also noted that non-Hispanic Asian race and ethnicity and marital status are associated with increased risk for MASLD. In agreement with previously published data, we noted that increased physical activity is associated with lower MASLD risk. Significant interactions between food security and other SDOH variables, including education level, suggest that targeting food-insecure groups for further study related to SDOH-based drivers of MASLD is warranted.

## Figures and Tables

**Figure 1 jcm-14-05484-f001:**
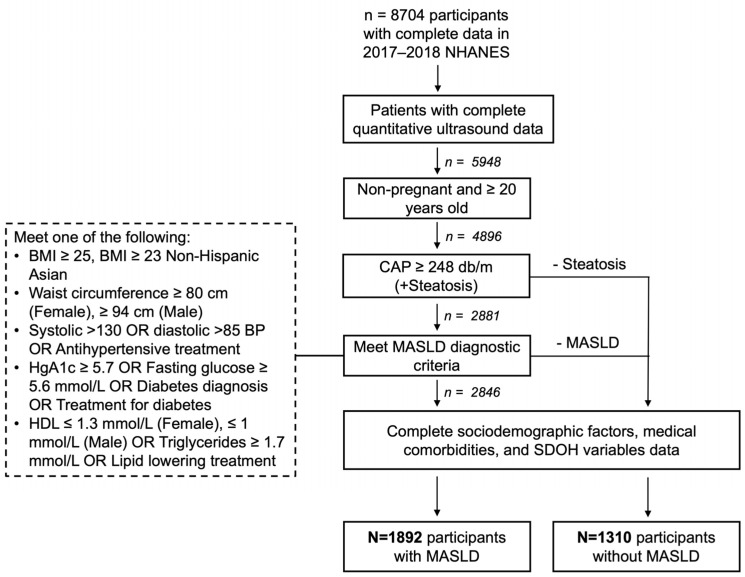
Study cohort inclusion and exclusion criteria. NHANES: National Health and Nutrition Examination Survey; CAP: controlled attenuation parameter; MASLD: metabolic dysfunction-associated steatotic liver disease; SDOH: social determinants of health; BMI: body mass index; HgA1c: hemoglobin A1c; HDL: high-density lipoprotein.

**Table 1 jcm-14-05484-t001:** Demographic and baseline characteristics of participants with and without metabolic dysfunction-associated steatotic liver disease (MASLD). BMI: body mass index.

	All Participants	Non-MASLD	MASLD	*p* Value *
Total [*n*]	3202	1310	1892	
Male [*n* (%)]	1644 (51.3)	607 (46.3)	1037 (54.8)	<0.001
Race/Ethnicity [*n* (%)]				<0.001
Non-Hispanic White	1239 (38.7)	507 (38.7)	732 (38.7)	
Mexican American	436 (13.6)	124 (9.5)	312 (16.5)	
Other Hispanic	269 (8.4)	103 (7.9)	166 (8.8)	
Non-Hispanic Black	722 (22.5)	349 (26.6)	373 (19.7)	
Non-Hispanic Asian	356 (11.1)	150 (11.5)	206 (10.9)	
Other Race/Multi-Racial	180 (5.6)	77 (5.9)	103 (5.4)	
Age [Mean (SD)]	50.6 (17.2)	47.0 (18.3)	53.2 (15.9)	<0.001
BMI [Mean (SD)]	29.9 (7.2)	26.01 (5.3)	32.61 (7.0)	<0.001
BMI Category (kg/m^2^) [*n* (%)]				<0.001
<18.5	756 (23.6)	580 (44.3)	176 (9.3)	
18.5–24.9	38 (1.2)	35 (2.7)	3 (0.2)	
25–29.9	1023 (31.9)	427 (32.6)	596 (31.5)	
≥30	1385 (43.3)	268 (20.5)	1117 (59.0)	
Comorbidities [*n* (%)]				
Hypertension	1208 (37.7)	341 (26.0)	867 (45.8)	<0.001
Hypercholesterolemia	1169 (36.5)	374 (28.5)	795 (42.0)	<0.001
Diabetes	475 (14.8)	93 (7.1)	382 (20.2)	<0.001
Steatosis Grade [*n* (%)]				<0.001
S0	1276 (39.9)	1276 (97.4)	0 (0.0)	
S1	321 (10.0)	16 (1.2)	305 (16.1)	
S2	239 (7.5)	3 (0.2)	236 (12.5)	
S3	1366 (42.7)	15 (1.1)	1351 (71.4)	
Fibrosis Grade [*n* (%)]				<0.001
F0 to F1	2638 (82.4)	1194 (91.1)	1444 (76.3)	
F2	337 (10.5)	83 (6.3)	254 (13.4)	
F3	124 (3.9)	16 (1.2)	108 (5.7)	
F4	103 (3.2)	17 (1.3)	86 (4.5)	
History of Hepatitis B [*n* (%)]	33 (1.0)	12 (0.9)	21 (1.1)	0.70
History of Hepatitis C [*n* (%)]	64 (2.0)	33 (2.5)	31 (1.6)	0.10
Alcohol Use [*n* (%)]				0.20
Never drink	724 (22.6)	275 (21.0)	449 (23.7)	
Drink daily	220 (6.9)	91 (6.9)	129 (6.8)	
Drink few times/week	692 (21.6)	289 (22.1)	403 (21.3)	
Drink few times/month	665 (20.8)	293 (22.4)	372 (19.7)	
Drink few times/year	901 (28.1)	362 (27.6)	539 (28.5)	

* *p* value compares MASLD versus non-MASLD groups.

**Table 2 jcm-14-05484-t002:** Social determinants of health characteristics of participants with and without metabolic dysfunction-associated steatotic liver disease (MASLD).

	All Participants	Non-MASLD	MASLD	*p* Value *
Total [*n*]	3202	1310	1892	
Income Level [*n* (%)]				0.69
<$65,000	1987 (62.1)	807 (61.6)	1180 (62.4)	
≥65,000	1215 (37.9)	503 (38.4)	712 (37.6)	
Education Level [*n* (%)]				<0.001
Less than college	2414 (75.4)	440 (72.1)	759 (77.8)	
College graduate or above	788 (24.6)	367 (27.9)	421 (22.2)	
Marital Status [*n* (%)]				<0.001
Single, widowed, never married, and/or separated	1297 (40.5)	596 (45.5)	701 (37.1)	
Married or living with partner	1905 (59.5)	714 (54.5)	1191 (62.9)	
Achieving Moderate Physical Recreational Activity [*n* (%)]	818 (25.5)	437 (33.2)	381 (20.1)	<0.001
Food Security [*n* (%)]				0.043
Marginal or low food security	1165 (36.3)	449 (34.3)	716 (37.8)	
Full food security	2037 (63.7)	861 (65.7)	1176 (62.2)	
Access to Healthcare Facility [n (%)]	2593 (81.0)	1017 (77.6)	1576 (83.3)	<0.001
Private Insurance [*n* (%)]	1658 (51.8)	659 (50.3)	999 (52.8)	0.18

* *p* value compares MASLD versus non-MASLD groups.

**Table 3 jcm-14-05484-t003:** Univariable and multivariable logistic regression model evaluating the association of education level and income bracket with metabolic dysfunction-associated steatotic liver disease (MASLD). OR: odds ratio; CI: confidence interval; BMI: body mass index.

	Univariable			Multivariable		
	OR	95% CI	*p* Value	OR	95% CI	*p* Value
Education						
Less than college	Ref					
College or above	0.74	0.63, 0.86	<0.001	0.77	0.62, 0.97	0.024
Income						
<65,000	Ref					
≥$65,000	0.97	0.84, 1.12	0.70	1.12	0.91, 1.37	0.3
Age	1.02	1.02, 1.03	<0.001	1.02	1.01, 1.03	<0.001
Male	1.4	1.22, 1.62	<0.001	1.42	1.19, 1.70	<0.001
Race/Ethnicity						
Non-Hispanic White	Ref					
Mexican American	1.74	1.38, 2.21	<0.001	1.71	1.29, 2.28	<0.001
Other Hispanic	1.12	0.85, 1.47	0.40	1.08	0.79, 1.49	0.6
Non-Hispanic Black	0.74	0.62, 0.89	0.001	0.63	0.50, 0.79	<0.001
Non-Hispanic Asian	0.95	0.75, 1.21	0.70	2.37	1.75, 3.23	<0.001
Other race	0.93	0.68, 1.27	0.60	1	0.69, 1.47	>0.9
Hypertension History	2.4	2.06, 2.80	<0.001	1.33	1.09, 1.62	0.006
Diabetes History	3.31	2.62, 4.22	<0.001	1.69	1.28, 2.25	<0.001
Hypercholesterolemia History	1.81	1.56, 2.11	<0.001	1.08	0.89, 1.32	0.4
BMI Category						
18.5–24.9	Ref					
<18.5	0.28	0.07, 0.80	0.037	0.4	0.10, 1.17	0.14
25–29.9	4.6	3.74, 5.68	<0.001	4.33	3.46, 5.46	<0.001
≥30	13.7	11.1, 17.1	<0.001	16.6	13.1, 21.3	<0.001
History of Hepatitis B	1.21	0.60, 2.55	0.60	0.93	0.41, 2.18	0.90
History of Hepatitis C	0.64	0.39, 1.06	0.08	0.57	0.31, 1.03	0.06
Alcohol Use						
Never drink	Ref					
Drink daily	0.87	0.64, 1.18	0.40	1.53	1.05, 2.23	0.026
Drink few times/week	0.85	0.69, 1.06	0.15	1.26	0.96, 1.64	0.09
Drink few times/month	0.78	0.63, 0.96	0.021	0.98	0.75, 1.28	0.90
Drink few times/year	0.91	0.75, 1.11	0.40	1.05	0.82, 1.34	0.70
Achieving Moderate Physical activity	0.5	0.43, 0.59	<0.001	0.72	0.59, 0.89	0.002
Full Food Security	0.86	0.74, 0.99	0.039	0.76	0.63, 0.93	0.007
Private Insurance	1.11	0.96, 1.27	0.20	1.19	0.99, 1.44	0.06
Access to Healthcare	1.44	1.20, 1.72	<0.001	1.01	0.80, 1.27	0.90
Married/Living with Partner	1.42	1.23, 1.64	<0.001	1.28	1.07, 1.53	0.008

**Table 4 jcm-14-05484-t004:** Stratified multivariable logistic regression model evaluating the association between education level and metabolic dysfunction-associated steatotic liver disease (MASLD) in participants with and without full food security. OR: odds ratio; CI: confidence interval; BMI: body mass index.

	Food-Secure Stratum			Food-Insecure Stratum		
	OR	95% CI	*p* Value	OR	95% CI	*p* Value
Education						
<College	Ref			Ref		
College or above	0.71	0.55, 0.91	0.007	1.26	0.76, 2.11	0.40
Age	1.02	1.01, 1.03	<0.001	1.02	1.01, 1.04	<0.001
Male	1.55	1.24, 1.93	<0.001	1.2	0.88, 1.64	0.30
Race/Ethnicity						
Non-Hispanic White	Ref			Ref		
Mexican American	1.83	1.26, 2.68	0.002	1.54	0.98, 2.44	0.06
Other Hispanic	0.99	0.65, 1.51	>0.90	1.11	0.67, 1.86	0.70
Non-Hispanic Black	0.72	0.54, 0.97	0.028	0.47	0.32, 0.69	<0.001
Non-Hispanic Asian	2.24	1.58, 3.19	<0.001	2.98	1.56, 5.86	0.001
Other race	1.05	0.64, 1.74	0.80	0.95	0.52, 1.76	0.90
Hypertension History	1.23	0.96, 1.58	0.10	1.57	1.10, 2.23	0.012
Diabetes History	1.96	1.38, 2.84	<0.001	1.22	0.77, 1.95	0.40
Hypercholesterolemia History	1.02	0.80, 1.29	0.90	1.2	0.85, 1.72	0.30
BMI Category						
18.5–24.9	Ref			Ref		
<18.5	0.42	0.07, 1.55	0.30	0.38	0.02, 2.05	0.40
25–29.9	3.89	2.95, 5.17	<0.001	5.63	3.77, 8.53	<0.001
≥30	15.7	11.6, 21.5	<0.001	19.8	13.1, 30.4	<0.001
History of Hepatitis B	1.33	0.49, 4.07	0.60	0.46	0.09, 2.16	0.30
History of Hepatitis C	0.72	0.30, 1.70	0.50	0.45	0.19, 1.09	0.07
Alcohol Use						
Never drink	Ref			Ref		
Drink daily	1.77	1.12, 2.83	0.015	1.21	0.63, 2.35	0.60
Drink few times/week	1.12	0.80, 1.56	0.50	1.62	1.02, 2.59	0.042
Drink few times/month	0.86	0.61, 1.20	0.40	1.23	0.79, 1.93	0.40
Drink few times/year	1.04	0.76, 1.42	0.80	1.01	0.68, 1.51	>0.9
Achieving Moderate Physical Activity	0.68	0.53, 0.87	0.002	0.86	0.58, 1.27	0.40
Private Insurance	1.18	0.93, 1.48	0.20	1.21	0.87, 1.69	0.30
Access to Healthcare	0.9	0.66, 1.22	0.50	1.14	0.79, 1.62	0.50
Married/Living with Partner	1.42	1.13, 1.79	0.003	1.16	0.86, 1.56	0.30

## Data Availability

2017–2018 NHANES study, available at https://wwwn.cdc.gov/nchs/nhanes/continuousnhanes/default.aspx?BeginYear=2017.

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
