# Peer review of "The Impact of Social Determinants of Health on Metabolic Dysfunction-Associated Steatotic Liver Disease Among Adults in the United States"

_jcm, 2025, doi:10.3390/jcm14155484_

Round 1
Reviewer 1 Report
Comments and Suggestions for Authors
The article is very interesting and addresses the growing global medical problem of MASLD, focusing on its prevalence in different national populations depending on educational, social, and economic factors.
The manuscript is well written, with clear tables and figures; the results and discussion are appropriate. The study’s limitations were mentioned, such as the lack of a distinct population with MetALD.
1.However, the manuscript does not mention limitations such as the absence of waist circumference measurements or information on patients with prediabetes.
2.There is also no analysis regarding the type and frequency of physical activity or the diet followed. It would be worthwhile to mention these points or add them in the discussion, especially in the context of the eating window and time-restricted eating (TRE).
3.Furthermore, there is no information about medications used by patients that could potentially affect the outcomes and the degree of steatosis, nor is there any data on cardiovascular disease history, which is one of the complications of MASLD.
4.It would also be valuable to include up-to-date references related to lifestyle medicine in the prevention and treatment of MASLD:
Rajewski, P.; Cieściński, J.; Rajewski, P.; Suwała, S.; Rajewska, A.; Potasz, M. Dietary Interventions and Physical Activity as Crucial Factors in the Prevention and Treatment of Metabolic Dysfunction-Associated Steatotic Liver Disease. Biomedicines 2025, 13, 217. https://doi.org/10.3390/biomedicines13010217
and The role of FibroScan in MASLD diagnosis:
Rajewski P, Ciescinski J, Rajewski P (2024) Use of Fibroscan Liver Elastography in the Rapid Diagnosis and Monitoring of MASLD Treatment. Ann Case Report. 9: 2129. https://doi.org/10.29011/2574-7754.102129
After addressing these comments, I believe the article can proceed to publication and may provide valuable guidance regarding education and care for specific patient groups at risk of developing MASLD.
Reviewer 2 Report
Comments and Suggestions for Authors
The objective of this manuscript was to identify participants with metabolic dysfunction-associated stetotic liver disease (MASLD) using liver ultrasound-based controlled attenuation parameter measurements that align with established diagnostic guidelines. Major modification is required.
Please remove all instances of ‘we’ and “our” from the manuscript, including the abstract. You may use phrases such as this study, current study, or present study instead of we or our.
The number of references is inadequate; there should be at least 50.
The abstract requires significant revision. The background section is too long. The objective stated in the methods section of the abstract should be moved to the background and objectives section. In the Methods section of the abstract, you should specify the type of study, the location, and the duration. There is no need to include details about statistical analysis in the abstract.
In the results section of the abstract, you should include the confidence interval (CI) following the odds ratio (OR).
The keywords should be updated based on relevant MeSH terms.
The introduction is too brief; it should be at least one page long.
The rationale for the study and the existing data gap should be explored and supported by relevant references before writing the objectives.
In the methods section, you should specify the type of study conducted. It appears that this was a retrospective cross-sectional study.
The conclusion section should be revised to include more key findings from the study. Please improve the discussion and compare the outcomes with other previous similar studies.
Extensive proofreading and paraphrasing are required.
Comments on the Quality of English LanguageExtensive proofreading and paraphrasing are required.
Reviewer 3 Report
Comments and Suggestions for Authors
This study investigates the relationship between social determinants of health (SDOH) and the prevalence of metabolic dysfunction-associated steatotic liver disease (MASLD) among U.S. adults, utilizing data from the 2017–2018 NHANES cycle. The authors focus particularly on education level and income, and explore their independent and interactive effects with other SDOH variables—especially food security—on the likelihood of MASLD. Logistic regression models are applied, and results suggest that higher educational attainment is associated with lower odds of MASLD, while income level shows no significant association. The study also finds that the protective effect of education is only present in individuals with full food security, highlighting a potential modifying role of food access in MASLD risk.
The research addresses an important and timely topic by examining how upstream social factors may influence metabolic liver disease outcomes. However, there are several methodological and interpretative concerns that limit the strength and generalizability of the findings. My specific comments are outlined below:
1. Binarization of Income
The decision to dichotomize household income using a $65,000 cutoff—although based on median U.S. income—may have led to an oversimplification of a complex socioeconomic variable. Such binarization reduces variability and may obscure non-linear associations with MASLD risk. This could partly explain why income was not found to be significantly associated with MASLD in your models.
Additionally, food security itself is likely influenced by income. Thus, the lack of association between income and MASLD might reflect measurement bias or unaccounted interactions. I recommend either modeling income as a continuous or ordinal variable, or performing sensitivity analyses using alternative categorizations.
2. Measurement of Food Security
The food security variable derived from NHANES is based on the U.S. Household Food Security Survey Module and reflects household-level, not individual-level, food access. This distinction is important when correlating food security with an individual health outcome such as MASLD. The limitations of using household-level data in this context should be more explicitly acknowledged in the discussion.
3. Inclusion of Alcohol Users Without Differentiation
The study does not exclude participants with moderate or high alcohol consumption, nor does it distinguish between MASLD and metabolic-alcoholic liver disease (MetALD). In a population like the U.S. where alcohol use is prevalent, this could substantially confound the results. Although the authors mention sensitivity analyses that excluded daily drinkers, more robust stratification or exclusion criteria for alcohol use would strengthen the validity of the MASLD definition.
4. Association Between Marital Status and MASLD
The observed association between being married/living with a partner and increased MASLD risk is intriguing, but insufficiently explored. Potential behavioral mechanisms—such as lower physical activity, dietary habits, or reduced health prioritization in married individuals—could be discussed. Discussions about the behavioral mechanisms underlying this issue (e.g., physical activity, body image perception, care behaviors) can be added to the literature. This could add valuable nuance to the interpretation of the findings, even if speculative.
My personal observation as a reviewer is that married individuals, especially those with children, may put themselves in the background, pay less attention to their health, and this may lead to physical inactivity; while single individuals may be more motivated in terms of appearance and health in order to be active in social interactions. This observation is not scientific, but it may contribute to the discussion.
5. Understated Limitations
Several important limitations deserve greater emphasis in the manuscript:
The cross-sectional nature of NHANES data limits causal inference.
Binarization of key variables (e.g., income, education) likely results in information loss.
The inability to distinguish MASLD from MetALD due to limited alcohol consumption data.
Food security measurement pertains to the household, not individuals.
Explicitly stating these limitations would improve transparency and contextualize the study’s conclusions more appropriately.
As a result, this is a well-structured and timely study that draws attention to the importance of education and food security in MASLD risk. However, some methodological decisions—especially the dichotomization of income and the absence of alcohol-related exclusions—may have significantly influenced the main findings. Clarifying these points and expanding the discussion around potential confounding factors and measurement limitations would enhance the manuscript’s rigor and interpretability.
Round 2
Reviewer 2 Report
Comments and Suggestions for Authors
The manuscript has low quality. It does not add anything to the subject area compared to other published Materials. So, any comments about the methodology are equivocal.
Reviewer 3 Report
Comments and Suggestions for Authors
The authors have successfully addressed the my comments and significantly improved the manuscript. While the study, by its nature, does not allow causal inferences, it provides valuable insights.